# Hot Air Drying of *Sipunculus nudus*: Effect of Microwave-Assisted Drying on Quality and Aroma

**DOI:** 10.3390/foods12040733

**Published:** 2023-02-08

**Authors:** Yaping Dai, Yupo Cao, Wei Zhou, Donghong Zhu

**Affiliations:** 1Key Laboratory of Tropical Crop Products Processing of Ministry of Agriculture and Rural Affairs, Agricultural Products Processing Research Institute, Chinese Academy of Tropical Agricultural Sciences, Zhanjiang 524001, China; 2Hainan Key Laboratory of Storage and Processing of Fruits and Vegetables, Zhanjiang 524001, China

**Keywords:** *Sipunculus nudus*, microwave pre-drying, nutritional quality, volatile components, hot air drying

## Abstract

The present study aimed to investigate the effect of different microwave pre-drying times under hot-air-drying processes on the quality properties and sensory evaluation of *Sipunculus nudus* (*S. nudus*)**.** The colour, proximate analysis, amino acid content, fat oxidation, and volatile components of dried *S. nudus* were determined. Microwave pre-drying could significantly (*p <* 0.05) increase the drying rate and shorten the drying time. The results of colour, proximate analysis, and amino acid content indicated that microwave pre-drying could improve the quality of the product, resulting in dried *S. nudus* with less of a loss in nutrients. The samples that underwent microwave pre-drying had a high degree of fatty acid oxidation and low monounsaturated fatty acid content, which facilitated the formation of volatile components. Additionally, the MAD-2 and MAD-3 groups had high relative contents of aldehydes and hydrocarbons, and the FD group had the highest relative content of esters found in the samples. The relative content of ketones and alcohols did not differ significantly between the different drying groups. The finding of this study has important potential for enhancing the quality and aroma of dry *S. nudus* products with microwave pre-drying during the drying process.

## 1. Introduction

The marine worm species *Sipunculus nudus (S. nudus),* which is a member of the phylum Sipuncula, has a body that resembles a section of intestine and is between 10 and 20 cm long [1]. *S. nudus* is well known as Marine Cordyceps Sinensis in some areas because it is rich in nutritional ingredients, including protein, polysaccharides, and fatty acids, which contribute to treating carbuncles, tuberculosis, and nocturia, regulating the function of the stomach and spleen, and treating disability and ageing caused by various pathogens [2]. In the previous research, more efforts were focused on the development of *S. nudus* in regard to harvesting and cultivation. *S. nudus* is rarely sold on the market because fresh samples have a short shelf-life caused mainly by the high moisture content (80–85%) of edible parts, autolytic enzyme, and lipid oxidation. Therefore, to extend its shelf-life, *S. nudus* can be processed into dried products for sale after being pre-treated. The dried form of aquatic food is widely consumed in Asian countries, including China and Japan [3]. Water is removed from the material after the drying process, resulting in a longer shelf-life for sales, less storage space, and lighter shipping weight. However, information on drying methods affecting the nutrition of *S. nudus* is scarce. Additionally, the typical drying process of *S. nudus* was unable to satisfy market demand for premium dehydrated goods.

Microwave drying is an effective method for the quality enhancement of dried products. The higher drying rate in microwave-vacuum drying is attributed to the promotion of moisture transfer from the interior of the drying sample to the surface, due to the microwave irradiation that penetrates the sample interior and selectively acts upon the contained water molecules [4]. Microwave-dried products provide better volatile substances, nutrition, and colour, and enhance the dehydration rate compared to traditional hot air drying [5]. Several studies have been conducted on microwave-related drying in the seafood industry. According to Brewer et al. [6], microwave heating could hasten the oxidation of lipids and myoglobin in foods containing muscle. According to the earlier reference, microwave pre-drying can increase the drying rate and product quality of apples and strawberries [7]. However, previous results have shown that single microwave drying might cause the degradation of nutrients and the occurrence of browning while microwave-assisted drying followed by hot air drying gains significant advantages compared with microwave drying alone. Different combined drying methods have a significant effect on the total phenolic and polysaccharide content of the dried products and their highest retention values of antioxidant activity [8]. In contrast, hot air drying without pre-drying could reduce the quality of meat products due to the oxidation of fatty acids during drying, leading to the browning of dried products.

Up until now, there has been a lack of research on the effect of pre-drying time on the quality of air-dried *S. nudus*. Herein, microwave pre-drying was carried out to reduce the oxidation of *S. nudus* meat in the air-drying process. The effect of microwave pre-drying time on changes in nutrients and volatile aroma substances in the hot-air dried *S. nudus* was investigated and compared with freeze-dried *S. nudus*.

## 2. Materials and Methods

### 2.1. Materials

Fresh *Sipunculus nudus* (*S. nudus*) was obtained from the Dongfeng market in Zhanjiang, Guangdong, China. The average weight of each *S. nudus* after removing visceral and coelomic fluid from its coelomic cavity was 3.00 ± 1.5 g. *S. nudus* coelomic cavity was boiled at 100 °C for 3 min to inhibit autolysis enzyme activity, followed by a drying experiment. The mean initial moisture content was 4.75 ± 0.2 (g/g d.b.). One batch of boiled *S. nudus* was utilised for all of the drying samples.

### 2.2. Drying Methods

Samples in the quantity of 100 g were placed in a single layer on a glassy culture dish and put into the microwave oven with a microwave power of 800 W (P70F20CL-DG(B0), Galanz Microwave Oven Electrical Appliance Co., Ltd. Foshan, China). Pre-drying times were set at 0, 1, 2, and 3 min and then they were dried via hot air drying at 55 °C with an air velocity of 1.5 m/s. The samples were denoted as MAD-0, MAD-1, MAD-2, and MAD-3 for the microwave pre-drying times of 0, 1, 2, and 3 min, respectively. The samples were weighed every 30 min until the moisture content of samples was below 5% (g/g, d.b.). For comparison, freeze-dried (FD) samples were also prepared via washing and sanitising *S. nudus*, and then frozen in a freeze dryer (Alpha 1-4 LDpius, Christ, Germany) for 45 h at −81 °C. The degree of vacuum was 0.37 Mpa throughout the experiments.

### 2.3. Moisture Content

The moisture content was determined using a moisture analyser (MRS 120-3, KERN, Balingen, Germany). For statistical purposes, the experiment was repeated three times and the mean value was used as the result.

### 2.4. Colour Evaluation

The surface colour of the samples was measured using a colorimeter (Color-5D, X-Rite Incorporation, Grand Rapids, MI, USA). The Hunter Lab colour scheme was utilised, where *L** stands for the luminance component’s lightness. Meanwhile, the *a** and *b** values represent the two chromatic components, where parameters range from green to red and from blue to yellow, respectively. Total colour difference (∆*E*) was calculated utilising the following expressions.
(1)$$\Delta E=\sqrt{{\left(L_{0}*-L*\right)}^{2}+{\left(a_{0}*-a*\right)}^{2}+{\left(b_{0}*-b*\right)}^{2}}$$
where, *L*_0_*, *a*_0_* and *b*_0_* were the value of the fresh sample, respectively; *L**, *a** and *b** represent the value of dried samples.

### 2.5. Proximate Analysis

The protein content was determined using Coomassie blue staining. Briefly, 0.5 g of the material was homogenised with 10 mL of distilled water in a 50 mL centrifugal tube. The solution was centrifuged at 1503× *g* for 15 min at 4 °C. A supernatant in the quantity of 1 mL was mixed with Coomassie Brilliant Blue solution (G-250) and allowed to form a chemical reaction for 2 min at 25 °C. The absorbance was measured at 595 nm using an ultraviolet–visible spectrophotometer (U-T6A, Yipu Instrument Manufacturing Co., Ltd., Shanghai, China) and the protein content was calculated using a standard reference of bovine serum albumin.

The total carbohydrate content was determined using the glucose calibration curve according to the modified method of Dubois et al. [9]. For the standard curve, 100 μg/mL of glucose was prepared. Serial dilutions of glucose (1 mL) were prepared and the same volumes of phenol (1 mL) and concentrated sulphuric acid (5 mL) were added to monitor the colour development at 470 nm using an ultraviolet–visible spectrophotometer (U-T6A, Yipu Instrument Manufacturing Co., Ltd., Shanghai, China). A standard curve was plotted and used to calculate the total carbohydrate concentration in the samples. The ash content was determined via the method proposed by Kardile et al. [10] using a muffle furnace (10-13A, Shanghai Kanglu Instrument and Equipment Co., Shanghai, China).

### 2.6. Amino Acid Analysis

The amino acid compositions were measured using an automatic amino acid analyser (S-433D, Sykam, Munich, Germany) according to the previous method with some modifications [11]. Briefly, 2 g of samples were hydrolysed in 6 M HCl-phenol solution for 24 h at 110 °C. Then, after centrifuging 1 mL of hydrolysate at 3381× *g* for 5 min, 200 L of the supernatant was evaporated using nitrogen blowing at 50 °C. To dissolve the remainder, 1.5 mL of 0.2 M HCl was added. The mixture was then filtered through a 0.22-μm-membrane filter. The amino acids were identified and quantified by comparing the peak profiles of the samples with standard amino acid profiles. Based on the amino acid composition, amino acid score (AAS) and essential amino acid index (EAAI) were analysed.

### 2.7. Fatty Acid Analysis

The fat was procured according to the hydrolysis extraction method of Cherifi et al. [12] with slight modifications. The fat should be esterified before being measured via gas chromatography (GC). Briefly, the samples were dissolved in 200 mL of potassium hydroxide methanol and 4 mL of *n*-hexane. The solution was shaken vigorously for 30 s and then left to stand until clarified. To balance the potassium hydroxide in the solution, 1 g of sodium bisulphate was added. After salt precipitation, the organic portion of the solution was taken out and passed through a 0.22 m organic membrane. The fatty acid was measured using a GC (GC 7820A/5925C MS, FRONTIER LAB Japan Agilent Technologies, Santa Clara, CA, USA). Nitrogen served as the carrier gas. The temperature was originally held at 100 °C for 13 min following injection (1.0 μL, split ratio: 100:1), and then during the analysis it was increased as follows: (1) to 180 °C at 10 °C/min and held for 6 min; (2) to 200 °C at 1 °C/min and held for 20 min; and (3) to 230 °C at 4 °C/min and held for 10.5 min. Triglyceride undecanoate was used as an internal standard substance. Fatty acids were calculated by comparing the relative response of the ratio of the internal standard with the relative response of the standard mixture.

### 2.8. Fat Oxidation

#### 2.8.1. Fat Extraction

According to the modified method [13], fat extraction was suitable for this determination. Briefly, 1 g of the sample was mixed with 2:1 chloroform–methanol solution (*v*/*v*). The mixture was blended via water bath vibration at 35 °C for 30 min. The solution was mixed with at least 5-fold its volume of distilled water (pH = 7.2), and centrifuged (Sigma 3–30K, Sigma Inc., Osterode am, Germany) at 3381× *g* for 5 min at 4 °C. The chloroform phase containing fat was transferred to a glass tubule and concentrated with nitrogen. The obtained fat was used to determine conjugated dienoic acid, *p*-Anisidine content, and fatty acid composition.

#### 2.8.2. Conjugated Dienoic Acid

According to the method of Lee et al. [14], 100 mg of fat was put into a 250 mL triangular bottle and added to 25 mL isooctane, then kept in the dark for about 10 min. The mixture was diluted 10 times with isooctane and the absorbance value (isooctane as blank) at 233 nm spectrum was measured using an ultraviolet–visible spectrophotometer (U-T6A, Yipu Instrument Manufacturing Co., Ltd., Shanghai, China). The results for each series of measurements were quantified according to the following equation.
(2)$$\mathrm{CDA}(\%)=\frac{0.84\times A_{233\mathrm{nm}}}{\mathrm{bc}-K_{0}}$$
where A_233 nm_ was the absorbance at 233 nm, b was the length of the cell (cm), c was the gram per litre of the sample, and K_0_ was the absorptivity by acid groups, with value 0.03.

#### 2.8.3. *p*-Anisidine Value Analysis

The *p*-AV of oxidised oils was determined by Xie et al. [15] with slight modification. The sample (100 mg) was mixed with 25 mL isooctane, and the absorbance of this mixture was measured at 350 nm using an ultraviolet–visible spectrophotometer (U-T6A, Yipu Instrument Manufacturing Co., Ltd., Shanghai, China). A quantity of 2.5 mL of the above mixture was mixed with 0.5 mL 0.5% (*w*/*v*) *p*-Anisidine, and the absorbance was read at 350 nm after 10 min in the dark. The *p*-AV of each sample was calculated using the following equation:(3)$$p-AV=\frac{25\times \left(1.2\times A_{2}-A_{1}\right)}{W}$$
where A_1_ was the absorbance before the addition of *p*-Anisidine at 350 nm, A_2_ was the absorbance after the addition of *p*-Anisidine at 350 nm, and W was the amount of sample in g.

### 2.9. Volatile Aroma Analysis

For all chromatographic studies, a 7890A-Gas-Chromatography/5977A-Mass-Spectrometer (GC-MS) system (GCMS-2010 Plus, Shimadzu, Kyoto, Japan) equipped with an Agilent HP-5-MS capillary column (30 m, 0.25 mm, 0.25 m) was used to separate the volatile aroma components of the samples. Initially maintained at 35 °C for 1 min, the temperature was then raised to 60 °C at 5 °C per minute and maintained for 1 min, 140 °C at 6 °C per minute and maintained for 1 min, and lastly 230 °C at 8 °C per minute and maintained for 5 min. The carrier gas was helium (>99.999%), flowing at a rate of 1.0 mL/min. Splitless sample injection was used, and the inlet port’s temperature was maintained at 250 °C. The mass spectrometry conditions were as follows: ionisation source temperature of 220 °C, ionisation energy of 70 eV, and full-scan mode in the mass range of 35–350 Amu. Utilising solid-phase microextraction, the volatile compounds were removed (SPME). As briefly stated, SPME was performed using 0.5 g of *S. nudus* powder, 9 mL of normal saline solution, and 30 L of 2-Octanol (10 g/mL), and moved into a vial. The SPME conditions were as follows: 60 °C for the extraction temperature, 40 min for the extraction time, and 15 min for the desorption time.

To verify the retention index, *n*-Alkane standards were measured under the same circumstances as the dried *S. nudus* samples. An automated mass spectral deconvolution and identification method was used to deconvolute the mass spectrum data to determine the co-elution peak. By contrasting the mass spectra (*m*/*z*) values of the retention index reported in the literature, the volatile chemicals were stably identified. Using 2-Octanol as an internal standard, semiquantitative analyses were carried out. Under the assumption that the response factor would be 1, the volatile chemical content was determined from the GC peak areas related to the GC peak area of the internal standard.

### 2.10. Statistical Analysis

All of the statistical analysis was conducted in triplicate. This means that SD was used to express the experimental outcomes. Origin 8.5 (2011, OriginLab Corporation, Northampton, MA, USA) and SPSS 16.0 (2010, IBM, Armonk, NY, USA) were used for the statistical analysis. ANOVA was used to evaluate the data (*p <* 0.05), and Duncan’s multiple range test was used to distinguish between the means.

## 3. Results and Discussion

### 3.1. Drying Characteristics

To analyse the effect of different microwave pre-drying times on the drying characteristics of *S. nudus*, the drying curves at different treatment times of 0, 1, 2, and 3 min (MAD-0, MAD-1, MAD-2, and MAD-3) were shown in Figure 1. As the microwave pre-drying times increased, the moisture content decreased (Figure 1A). The moisture content of MAD-0 and MAD-1 was reduced to less than 5% (g/g drying, d.b.) after 270 min. Compared to MAD-0 and MAD-1, the reduction in drying time was 33.3% and 44.4% for MAD-2 and MAD-3, respectively. The results reported in the present study are in agreement with the result previously reported by Song et al. [8]. The moisture content of MAD-0 and MAD-1 samples decreased significantly in the first 120 min of the drying process, but the change was not significant after continual drying. This could be due to the rapid mass transfer caused by microwave radiation, which would cause the product structure to change due to “puffing”, or its porous structure. This might be beneficial to the evaporation of water in the later hot-air-drying process. Subsequently, with the increase in drying time, the surface of the sample hardened, which hindered the evaporation of water in the later drying period [16]. From Figure 1B, the variations in drying rate under various microwave pre-drying times are shown. The drying rate exhibited a trend of increasing and then decreasing drying rates. The initial rate of increase could be because microwave pre-drying helped to enhance the reaction temperature and increase the driving force of heat and mass transfer which possess high air drying rates [17]. Due to the internal heat and mass transfer resistance, the drying rate subsequently decreased in the later stages of drying.

### 3.2. Colour

The colour parameters of air-dried *S. nudus* subjected to the different drying processes are listed in Table 1. The microwave pre-dried group showed significantly lower (*p <* 0.05) *L** values and significantly higher *a** and *b** values (*p <* 0.05) compared to the fresh samples, which might be attributable to the presence of Maillard reaction products during microwave pre-drying at high temperatures [18]. The sample dried via FD and MAD-3 groups showed higher *L** values compared to the MAD-0 and MAD-1 groups. This suggests that the FD and MAD-3 groups could increase the lightness, which might be due to the shorter drying time of the MAD-3 group and the lower drying temperature of the FD. Additionally, the *b** value (blueness at negative and yellowness at positive) was the important indicator for the browning. With the increase in microwave pre-drying time, the values of *a** and *b** initially increased and then decreased. These results show that the samples were drastically browned during thermal drying. Moreover, the total colour change (∆*E*) was significantly higher (*p <* 0.05) in the MAD group than in the freeze-dried group. It was noted that the value of ∆*E* of MAD-3 was significantly lower (*p <* 0.05) than that of other groups. The results might be ascribed to non-enzymatic browning reactions [19].

### 3.3. Proximate Analysis

In this section, we analyse the proximate composition to judge the nutritional value of dried products through different drying processes. As shown in Table 1, the moisture content of dried samples in three drying groups was lower than 5%, inhibiting microbiological growth and retarding enzymatic activity [20]. The protein content, total carbohydrate content, and ash content of the freeze-dried *S. nudus* were slightly higher than that of the microwave pre-drying groups. This might be related to protein denaturation and browning reactions with microwave pre-drying heating. Additionally, the longer the microwave pre-drying time, the higher the protein content and total carbohydrate content of dried *S. nudus*. The enzymatic processes caused proteins to partially degrade and change their structural makeup, revealing additional hydrophobic zones. As a result, there were more detectable proteins present. These results indicate that the retention of total carbohydrate, protein, and ash content was affected by different microwave pre-drying times. When the microwave pre-drying time was 3 min, the result of each component was close to the freeze-drying group. This means that a microwave pre-drying time of 3 min is suggested for the optimised processing of *S. nudus*.

### 3.4. Amino Acid Analysis

Amino acids are an important indicator of seafood quality, which is mainly influenced by the type of amino acid and the processing method [21]. The compositions of amino acids in all groups are reported in Table 2. The main amino acids of the samples were found to be glutamic acid, aspartic acid, arginine, leucine, and lysine, and the less abundant ones were cystine, phenylalanine, and methionine. Lysine and leucine were found to be the most abundant essential amino acids (EAAs) in dried samples, averaging higher than 65 mg/g. The glutamic acid also constituted the highest amount of non-essential amino acids (NEAAs) (125.30–147.75 mg/g). The MAD-3 group had the lowest total essential amino acid content, especially where the content of all essential amino acids was significantly (*p <* 0.05) lower than the other dried samples. The MAD-2 group had the highest total essential amino acids at 309.95 mg/g, which was 14.34% higher than the MAD-3 group. The amino acid content of the MAD-0, MAD-1, and MAD-2 groups was not significantly different (*p >* 0.05), and was similar to the content reported for grass carp fillets [22]. For non-essential amino acids, the MAD-3 group had significantly lower levels of cystine than the other drying groups, while the opposite was true for histidine. This phenomenon demonstrates the thermal stability of histidine and cystine. The amino acid composition of MAD-3 was significantly lower than that of the FD group. This was due to the degradation of nitrogenous compounds via heat treatment [23]. Moreover, the AAS according to the FAO/WHO standard model is shown in Table 2. It can be seen that the limiting amino acid in the MAD-0 group was valine, while MAD-1, MAD-2, MAD-3, and FD had limited valine as well as methionine and cystine. In addition, the essential amino acid index (EAAI) of MAD-0, MAD-1, MAD-2, MAD-3, and FD was 85.919, 83.41, 86.15, 73.97, and 78.70, respectively. The EAAI values were slightly higher for 2 min of microwave pre-drying, while the lowest values were found for 3 min of microwave pre-drying. This might show that prolonged high heat treatment causes water loss and removes some amino acids in *S. nudus*.

### 3.5. Fatty Acid Composition

Figure 2 shows the content of fatty acids with different drying methods. The dried *S. nudus* contained 25 fatty acids, with the total content of unsaturated fatty acids (UFAs) being higher than that of saturated fatty acids (SFAs). Myristic acid (C14:0), palmitic acid (C16:0), and stearic acid (C18:0) were the main components of the SFAs. Oleic acid (C18:1) and cis-11-Eicosenoic acid (20:1) were the main components of saturated monounsaturated fatty acids (MUFAs). Linoleic acid (C18:2), arachidonic acid (C20:4), and eicosapentaenoic acid (C20:5) were the main PUFAs. The content of SFAs and MUFAs in the dried *S. nudus* increased slightly with the increase in microwave pre-drying time, while the content of UFAs and PUFAs slightly decreased. However, studies have shown that the higher the UFAs, the less oxidative stability. Moreover, the SFA, UFA, monounsaturated fatty acid (MUFA), and polyunsaturated fatty acid (PUFA) content of freeze-dried samples were 29.626, 70.37, 26.820, and 43.552, respectively. In the MAD-3 group, the total SAF and MUFA content increased by 5.6% and 5.5%, respectively, and the total PUFA content decreased by 7.4% compared to the FD group. This might be because microwave volume heating destroys unsaturated fatty acids [24]. Therefore, microwave pre-drying was beneficial for improving the oxidative stability of samples. Lower levels of polyunsaturated fatty acids in dried samples could contribute to reducing the risk of oxidation since polyunsaturated fatty acids are substrates for oxidation [25].

### 3.6. Fat Oxidation

The conjugated dienoic acid value and *p*-anisidine value are important indicators for evaluating the quality of fat. They are used to measure the degree of oxidation and decomposition of polyunsaturated fatty acids and to detect secondary oxidation products (such as aldehydes) [26]. The conjugated dienoic acid and *p*-anisidine values of *S. nudus* are shown in Figure 3. The conjugated dienoic acid values of MAD-0, MAD-1, MAD-2, MAD-3, and FD were about 0.14, 0.16, 0.14, 0.20, and 0.19, respectively. The MAD-3 group had the highest value of conjugated dienoic acid and the MAD-0 group had the lowest value, which indicates that microwave pre-drying increases the fat oxidation of *S.nudus*. This might be because fatty acids are easily oxidised under light, heat, and oxygen conditions [27]. On the other hand, the *p*-anisidine value was related to the trend of the conjugated dienoic acid value. For the MAD-2 and MAD-3 groups, the *p*-anisidine value was significantly higher (*p <* 0.05) than in other groups. Similar results were reported by Jeon et al. [28] for roasting mealworm oil.

### 3.7. Volatile Components

The aroma characteristics of dried products play an important role in the acceptance and preference of dried products. The volatile components of dried *S.nudus* prepared via different drying methods are shown in Figure 4A–C. The number of compounds detected in MAD-0, MAD-1, MAD-2, MAD-3, and FD groups was 38, 37, 39, 44, and 30, respectively, and the content was 2.5765, 3.2784, 8.9892, 9.4200, and 15.7552 µg/g for aroma compounds. There were significant differences in volatile characteristics among different samples, indicating that the drying process affected the volatile components of *S.nudus*. The principal volatile compounds in dried *S.nudus* are aldehydes, alcohols, aromatics, hydrocarbons, esters, ketones, and nitrogenous compounds, which are also reported in sea cucumber [29]. From Figure 4A, the relative content of esters is seen to be higher in the samples dried via the FD method, while the relative content of esters is seen to be lower in the dried naked algae prepared in the MAD group. In contrast, for MAD-2 and MAD-3, the relative content of aldehydes and hydrocarbons was higher than that of other drying methods. The highest relative content of alcohols was discovered in samples in the FD group, followed by MAD-2. Additionally, there was no significant difference in the relative contents of ketones and alcohols among different drying groups.

The top five volatile compounds in MAD-0 groups were 2,3-Octanedione and 2-Octanone (0.4340 and 0.1216 µg/g sample, ketones with fruity aroma), nonanal (0.2481 µg/g sample, aldehydes with grass-fatty aroma), 1,2-dimethyl-Benzene (0.1514 µg/g sample, fragrance of flowers), and 1-Hexadecanol (0.1379 µg/g sample, delicate aroma). For MAD-1 groups, the top five volatile compounds were 2,3-Octanedione (0.3361 µg/g sample, ketones with fruity aroma), benzene, 1,2-dimethyl- and 1-Octen-3-ol (0.3011 and 0.2395 µg/g sample, the fragrance of flowers), nonanal (0.2520 µg/g sample, aldehydes with grass-fatty aroma), and 1-Hexanol, 2-ethyl- (0.1996 µg/g sample, delicate aroma). The top five volatile compounds in MAD-2 groups were Oxime-, methoxy-phenyl- (3.3915 µg/g sample, nitrogenous compounds), nonanal (0.3713 µg/g sample, aldehydes with grass-fatty aroma), 2,3-Octanedione (0.3038 µg/g sample, ketones with fruity aroma), 5,9-Undecadien-2-one, and 6,10-dimethyl- and 2-Tridecanone (0.2806 and 0.2719 µg/g sample, mushroom soil). The top five volatile compounds in MAD-3 groups were Oxime-, methoxy-phenyl- (3.0771 µg/g sample, nitrogenous compounds), nonanal (0.5576 µg/g sample, aldehydes with grass-fatty aroma), pentadecane (0.6258 µg/g sample, hydrocarbons), 2,3-Octanedione (0.4450 µg/g sample, ketones with fruity aroma), and 2-Tridecanone (0.2809 µg/g sample, mushroom soil). As for the FD groups, the top five volatile compounds were 2,2-Dimethyl-N-phenethylpropionamide (4.0625 µg/g sample, nitrogenous compounds), Oxime-, methoxy-phenyl- (1.4442µg/g sample, nitrogenous compounds), 2,5-Dihydroxyphenylacetic acid, ethyl ester, di-TMS (3.6697 µg/g sample, esters), 2-Octanone (0.6913µg/g sample, ketones), and azulene (1.0905 µg/g sample, hydrocarbons) [30].

The overall effects of different drying methods on the volatile compositions of *S. nudus* were analysed using principal component analysis (PCA). PC1 and PC2 were retained on the basis of the cumulative percentage of total variation (98.1%). The first two principal components accounted for 62.1% and 36.0% of the total variance, respectively. Figure 4C showed the score and loading bioplots defined by the first two PCs obtained from the PCA. The volatile aroma compounds in the dried products of *S. nudus* obtained via different drying methods were significantly different according to their PCA score plots. The MAD-0, MAD-1, MAD-2, and MAD-3 groups were placed in the negative PC1 direction while FD was placed in the positive direction. In addition, the groups of MAD-0, MAD-1, and FD were located in the same negative direction as PC2. The relatively low content of compounds was positively relevant to this axis [31]. Alcohols, esters, and aldehydes contributed to a large extent to PC1. Aldehydes, hydrocarbons, and aromatics contributed to a large extent to PC2.

## 4. Conclusions

In this study, the effects of microwave pre-drying for different times before the traditional hot-air-drying process on the nutritional components and volatile aroma of dried *Sipunculu snudus* (*S. nudus*) were investigated. Microwave pre-drying at 3 min (MAD-3) shortens the drying time by 44.4% compared to pre-drying for 0 min (MAD-0) (similar to the FD drying group). The amino acid composition and fatty acid composition of microwave pre-drying for 2 min had the most obvious advantages in all drying groups. Microwave pre-drying for 3 min showed a low content of MUFA (28.38 g/100 g dry basis). As demonstrated in this study, microwave pre-drying could help to reduce nutritional losses in dried products. The GC-MS results showed a gradual increase in the total amount of volatile compounds with increased microwave pre-drying time. Overall, microwave pre-drying could improve the quality and aroma of dried *S. nudus*, as compared with conventional hot air drying. Therefore, microwave pre-drying is worth considering in the processing of *S. nudus* in terms of production costs.

## Figures and Tables

**Figure 1 foods-12-00733-f001:**
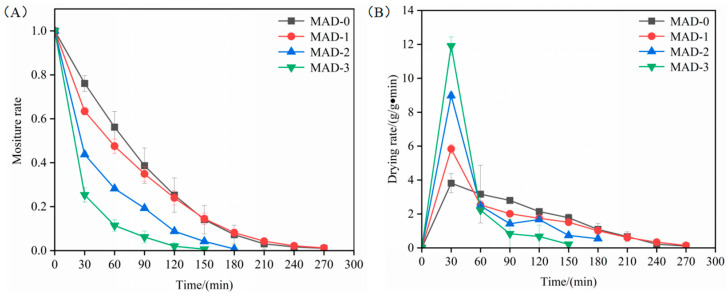
Effects of different forms of microwave pre-drying on drying characteristics for hot air drying *S. nudus.* (**A**): drying curves; (**B**): drying rate curves.

**Figure 2 foods-12-00733-f002:**
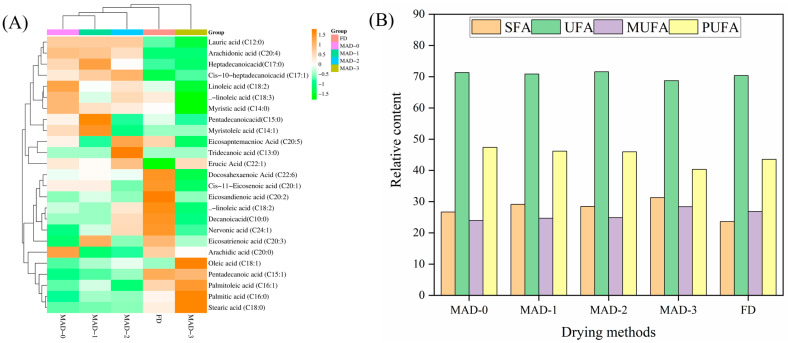
Fatty acid composition (**A**,**B**) of different microwave pre-drying methods in air-dried *S. nudus*.

**Figure 3 foods-12-00733-f003:**
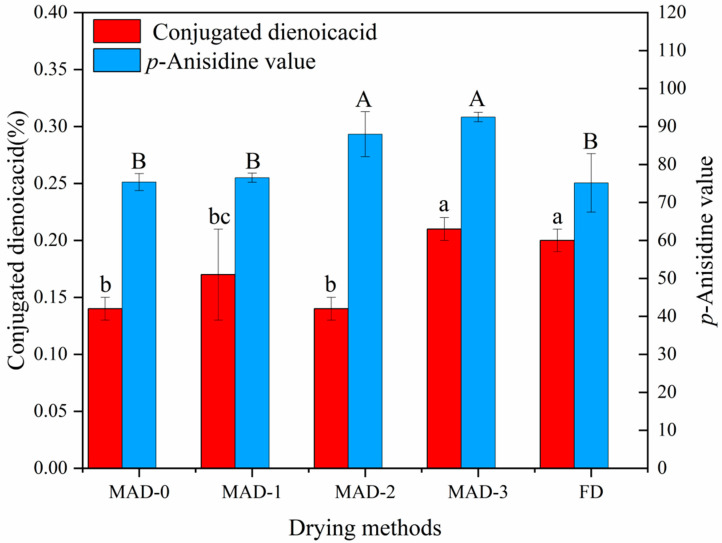
Conjugated dienoic acid and *p*-Anisidine values of air-dried *S. nudus* with different microwave pre-drying methods. Different letters in the Figure means significance difference (*p* < 0.05).

**Figure 4 foods-12-00733-f004:**
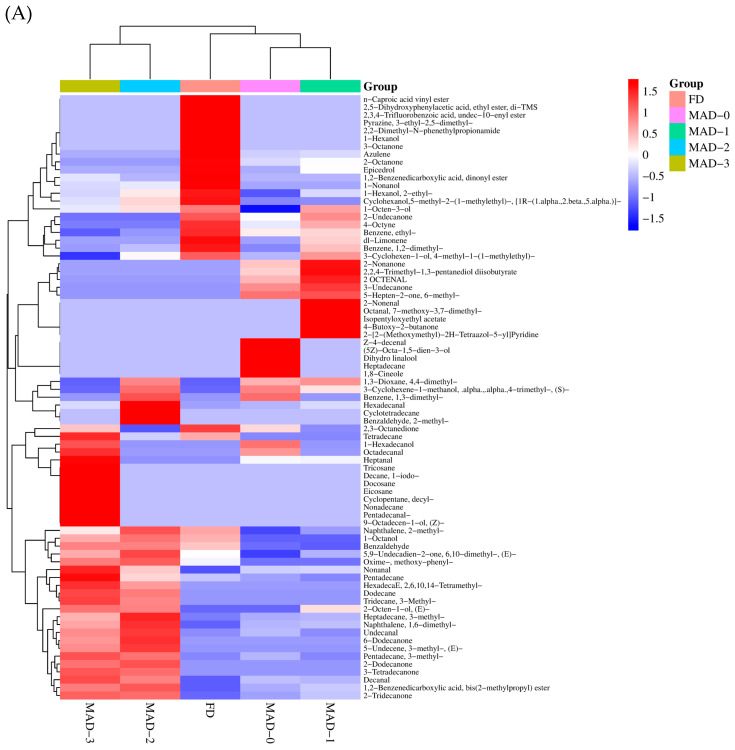
Effects of different microwave pre-drying methods on the contents of volatile aroma components (**A**,**B**), and score and loading plots (**C**) from the principal component analysis of dried *S. nudus*.

**Table 1 foods-12-00733-t001:** Comparison of colour and proximate analysis of dried *S. nudus* with different forms of microwave pre-drying.

	Parameters	Fresh	MAD-0	MAD-1	MAD-2	MAD-3	FD
Colour	*L**	54.13 ± 3.49 ^b^	39.72 ± 2.49 ^d^	42.09 ± 5.45 ^d^	43.86 ± 4.07 ^d^	48.13 ± 3.03 ^c^	69.42 ± 3.43 ^a^
	*a**	−0.06 ± 0.97 ^b^	1.69 ± 0.63 ^a^	2.1 ± 0.83 ^a^	1.25 ± 0.94 ^a^	0.68 ± 0.96 ^a^	−0.23 ± 0.25 ^c^
	*b**	−5.49 ± 1.23 ^c^	15.59 ± 2.22 ^a^	17.93 ± 2.59 ^a^	17.76 ± 2.54 ^a^	17.66 ± 3.86 ^a^	9.73 ± 1.36 ^b^
	Δ*E*	--	25.59 ± 4.50 ^a^	26.42 ± 3.69 ^a^	25.48 ± 3.39 ^a^	23.92 ± 4.02 ^b^	21.58 ± 3.07 ^c^
Proximate Analysis	Moisture content (g/100 g, d.b.)	--	4.70 ± 0.16 ^a^	3.86 ± 0.48 ^a^	3.65 ± 0.21 ^a^	2.96 ± 0.07 ^a^	2.96 ± 0.07 ^a^
	Protein content (mg/g)	--	33.45 ± 1.36 ^d^	33.77 ± 0.57 ^d^	39.83 ± 0.20 ^c^	44.96 ± 0.37 ^b^	56.23 ± 0.33 ^a^
	Total carbohydrate content (g/100 g)	--	1.14 ± 0.01 ^c^	1.28 ± 0.01 ^c^	1.28 ± 0.01 ^c^	1.69 ± 0.01 ^b^	3.04 ± 0.02 ^a^
	Ash content (g/100 g)	--	10.12 ± 0.56 ^a^	9.98 ± 0.23 ^a^	9.89 ± 0.33 ^a^	10.02 ± 0.26 ^a^	10.56 ± 0.22 ^a^

The data are shown as mean ± SD (n = 3). Significant differences (*p <* 0.05) are indicated by different letters in the same column. MAD-0, microwave pre-drying 0 min; MAD-1, microwave pre-drying 1 min; MAD-2, microwave pre-drying 2 min; MAD-3, microwave pre-drying 3 min; FD, freeze-drying.

**Table 2 foods-12-00733-t002:** Amino acid composition and the amino acid score of different dried *S. nudus*.

Amino Acids		Content (mg/g)				
		MAD-0	MAD-1	MAD-2	MAD-3	FD
Essential amino acids (EAAs)						
Lysine		67.55 ± 0.21 ^a^	66.30 ± 0.42 ^b^	68.00 ± 1.27 ^a^	58.80 ± 1.70 ^c^	65.70 ± 4.24 ^b^
Phenylalanine		27.85 ± 0.92 ^b^	29.30 ± 1.13 ^a^	30.05 ± 0.78 ^a^	25.00 ± 1.27 ^c^	27.25 ± 2.05 ^b^
Threonine		42.7 ± 0.71 ^a^	40.4 ± 0.57 ^b^	42.05 ± 1.48 ^a^	36.30 ± 1.13 ^c^	40.40 ± 2.97 ^b^
Isoleucine		37.85 ± 0.49 ^b^	35.65 ± 0.64 ^c^	38.00 ± 0.99 ^a^	32.60 ± 0.99 ^d^	35.50 ± 2.40 ^c^
Leucine		69.45 ± 0.78 ^a^	65.7 ± 0.99 ^b^	69.50 ± 1.70 ^a^	59.70 ± 1.98 ^c^	65.30 ± 4.81 ^b^
Valine		36.95 ± 0.21 ^a^	34.65 ± 1.06 ^b^	36.75 ± 1.06 ^a^	31.55 ± 0.92 ^c^	34.85 ± 1.91 ^b^
Methionine		26.75 ± 0.35 ^a^	21.65 ± 0.21 ^b^	25.60 ± 0.00 ^a^	21.55 ± 0.49 ^b^	19.95 ± 1.06 ^b^
Total essential amino acids (E)		309.1 ± 3.68 ^a^	293.65 ± 4.60 ^a^	309.95 ± 5.73 ^a^	265.50 ± 7.50 ^c^	288.95 ± 17.32 ^b^
Non-essential amino acids (NEAAs)						
Tyrosine		31.75 ± 0.49 ^a^	29.95 ± 0.35 ^b^	32.10 ± 0.57 ^a^	27.35 ± 0.64 ^c^	29.80 ± 2.26 ^b^
Cystine		4.50 ± 0.14 ^b^	5.05 ± 0.78 ^a^	3.40 ± 0.28 ^c^	4.25 ± 0.21 ^b^	3.20 ± 1.84 ^c^
Histidine		22.65 ± 0.07 ^b^	21.85 ± 0.49 ^b^	22.40 ± 0.57 ^b^	19.30 ± 0.14 ^c^	24.65 ± 0.35 ^a^
Glutamic acid		147.75 ± 1.91 ^a^	141.20 ± 2.83 ^b^	145.05 ± 5.02 ^a^	125.30 ± 4.38 ^c^	140.35 ± 9.97 ^b^
Aspartic acid		93.55 ± 0.49 ^a^	88.65 ± 1.20 ^b^	92.35 ± 2.62 ^a^	80.00 ± 2.97 ^c^	91.45 ± 6.29 ^a^
Arginine		88.75 ± 1.06 ^b^	86.30 ± 3.68 ^b^	88.15 ± 0.64 ^b^	75.15 ± 2.90 ^c^	93.60 ± 3.96 ^a^
Alanine		63.60 ± 1.41 ^a^	59.70 ± 1.13 ^b^	61.65 ± 2.47 ^a^	53.80 ± 1.84 ^c^	61.35 ± 4.31 ^a^
Glycine		64.70 ± 2.55 ^b^	59.80 ± 1.27 ^c^	60.40 ± 3.11 ^c^	52.65 ± 2.05 ^d^	68.40 ± 4.10 ^a^
Serine		37.10 ± 0.71 ^a^	35.30 ± 0.71 ^a^	36.55 ± 1.20 ^a^	31.50 ± 1.13 ^b^	35.30 ± 2.55 ^a^
Proline		32.05 ± 0.21 ^a^	30.65 ± 0.07 ^a^	31.50 ± 1.98 ^a^	28.00 ± 0.42 ^b^	30.45 ± 2.47 ^a^
Total non-essential amino acids (N)		586.40 ± 6.79 ^a^	558.45 ± 12.37 ^a^	573.55 ± 17.32 ^a^	497.30 ± 16.69 ^c^	578.55 ± 38.11 ^a^
Total amino acid E/N ratio		0.527 ± 0.0001 ^b^	0.526 ± 0.003 ^b^	0.540 ± 0.006 ^a^	0.533 ± 0.002 ^b^	0.499 ± 0.002 ^c^
Index						
AAS	Threonine	128.32 ± 2.12 ^a^	125.11 ± 1.75 ^a^	127.48 ± 4.50 ^a^	109.61 ± 3.42 ^b^	121.37 ± 8.92 ^a^
	Valine	89.55 ± 0.51 ^a^	86.53 ± 2.65 ^b^	89.85 ± 2.59 ^a^	76.83 ± 2.24 ^c^	84.43 ± 4.63 ^b^
	Methionine + Cystine	106.71 ± 1.69 ^a^	93.96 ± 1.99 ^b^	99.90 ± 0.97 ^b^	88.52 ± 0.97 ^c^	79.03 ± 2.66 ^d^
	Isoleucine	113.74 ± 1.49 ^a^	110.40 ± 1.97 ^a^	115.20 ± 3.00 ^a^	98.43 ± 2.99 ^c^	106.65 ± 7.22 ^b^
	Leucine	118.58 ± 1.33 ^a^	115.60 ± 1.74 ^a^	119.71 ± 2.92 ^a^	102.42 ± 3.40 ^b^	111.46 ± 8.20 ^a^
	Phenylalanine + Tyrosine	117.83 ± 2.80 ^b^	120.71 ± 3.03 ^a^	123.95 ± 0.42 ^a^	103.99 ± 3.79 ^c^	112.75 ± 8.52 ^b^
	Lysine	149.26 ± 0.47 ^a^	150.97 ± 0.97 ^a^	151.58 ± 2.84 ^a^	130.55 ± 3.77 ^c^	145.12 ± 9.37 ^b^
EAAI		85.92 ± 1.10 ^b^	83.41 ± 1.55 ^b^	86.15 ± 1.72 ^a^	73.97 ± 1.88 ^d^	78.70 ± 5.00 ^c^

AAS: amino acid score; EAAI: essential amino acid index. MAD-0, microwave pre-drying 0 min; MAD-1, microwave pre-drying 1 min; MAD-2, microwave pre-drying 2 min; MAD-3, microwave pre-drying 3 min; FD, freeze-drying. Different letters in the Table means significance difference (*p* < 0.05).

## Data Availability

Data is contained within the article.

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
