# Peer review of "Hot Air Drying of Sipunculus nudus: Effect of Microwave-Assisted Drying on Quality and Aroma"

_foods, 2023, doi:10.3390/foods12040733_

Round 1

Reviewer 1 Report

Sipunculus nudus has high nutritional value. Development of novel technologies for its processing has high relevance. Drying is one of the key steps in processing. Microwave pre-drying (before conventional drying method) can be a viable option. Therefore, the topic of the manuscript can be interesting for the readers and can provide useful data for the industry, respectively. In the research, special focus was on the effects of preliminary microwave treatment on drying characteristics, colour, change in product components (fatty acids, amino acids etc.). The manuscript has a proper structure. Introduction section summarizes well the relevance of the study. Materials and methods are described clearly and in details. Manuscript contains interesting and valuable results that has practical relevance, as well. Results are discussed in details with relevant references. In my opinion, the manuscript has high scientific quality.

Comments, suggestions:

Please describe briefly in the Introduction section as well, how affect the microwave irradiation to achieve enhanced drying efficiency (high temperature ramp cause cell wall destruction that manifested in better dehydration, for instance?).

Figure 1 has low quality. Please improve the visibility of this figure.

Please check the typos in the mnauscript (ref, numbering; ‘aromaof’ in line 384).

Author Response

Response to Reviewer 1 Comments

Point 1: Please describe briefly in the Introduction section as well, how affect the microwave irradiation to achieve enhanced drying efficiency (high temperature ramp cause cell wall destruction that manifested in better dehydration, for instance?).

Response 1: Thank you very much for your circumspection. We have added some describe about microwave irradiation in the Introduction, and the detailed revision could be found in the revised version. (Please see Line 48-51)

Point 2: Figure 1 has low quality. Please improve the visibility of this figure.

Response 2: Thanks for the referee’s kind advice. Figure 1 has been revised in the revised version. (Please see Line 239)

Point 3: Please check the typos in the manuscript (ref, numbering; ‘aromaof’ in line 384).

Response 3: Thanks for the referee’s kind advice. We have checked the typos in the manuscript, and the detailed revision could be found in line 81, 399, and 418.

Reviewer 2 Report

This work investigated the effect of different microwave pre-drying times under hot-air drying processing on the quality properties and sensory evaluation of Sipunculus nudus.

I would like to suggest some comments listed below.

The page numbering is incorrect, since after page 6, the numbering starts again, please change it.

Page 1: Abstract.

Lines 11-12: I suggest complete this sentence as follow: The present study aimed to investigate the effect of different microwave pre-drying times under hot-air drying processing on the quality properties and sensory evaluation of Sipunculus nudus.

Line 27: The reference must be cited before adding the period at the end of the sentence, as follow: It is widely cultivated in coastal areas all over the world [1]. In this case, all manuscript needs to be revised.

Page 6: Figure 1: It is necessary to include and explain letters A) and B) in this figure legend.

Page 7: Table 1 legend: Add space after “Table” and “analysis”.

Page 8: Table 2: Insert a line above “Essential amino acids (EAA)” and “mg/g protein”.

Page 10: Figure 2: It is also necessary to include and explain letters A) and B) in this figure legend. Besides it is not possible to understand the results shown in this figure 2A, please increase its resolution.

Page 12: Figure 4: The same comments made in Figure 2 are apply to Figure 4.

Author Response

Response to Reviewer 2 Comments

Point 1: The page numbering is incorrect, since after page 6, the numbering starts again, please change it.

Response 1: Thank you very much for your circumspection. The page numbering has been changed in the revised version.

Point 2: Page 1: Lines 11-12: I suggest complete this sentence as follow: The present study aimed to investigate the effect of different microwave pre-drying times under hot-air drying processing on the quality properties and sensory evaluation of Sipunculus nudus.

Response 2: Thanks for the referee’s kind advice. We have replaced the sentence in the manuscript. (Please see Line 12-14)

Point 3: Page 1: Line 27: The reference must be cited before adding the period at the end of the sentence, as follow: It is widely cultivated in coastal areas all over the world [1]. In this case, all manuscript needs to be revised.

Response 3: Thanks for the referee’s kind advice. The format of citation in all manuscript has been modified in the revised version.

Point 4: Page 6: Figure 1: It is necessary to include and explain letters A) and B) in this figure legend.

Response 4: Thank you very much for your circumspection. We have added the explain letters A) and B) in this figure legend in the revised version.

Point 5: Page 7: Table 1 legend: Add space after “Table” and “analysis”

Response 5: Thank you very much for your circumspection. We have added a space after “Table” and “analysis”in Table 1 legend. (Please see Line 277)

Point 6: Page 8: Table 2: Insert a line above “Essential amino acids (EAA)” and “mg/g protein”.

Response 6: Thank you very much for your circumspection. Table 2 format has been modified in the revised version.

Point 7: Page 10: Figure 2: It is also necessary to include and explain letters A) and B) in this figure legend. Besides it is not possible to understand the results shown in this figure 2A, please increase its resolution.

Response 7: Thanks for the referee’s kind advice. We have modified the figure legend and resolution of Figure 2, and the detailed could be found in the revised version.

Point 8: Page 12: Figure 4: The same comments made in Figure 2 are apply to Figure 4.

Response 8: Thanks for the referee’s kind advice. We have modified the figure legend and resolution of Figure 4, and the detailed could be found in the revised version.

Reviewer 3 Report

Revise the title, perhaps it would be better to talk about microwave pre-treatment or assisted drying.

If possible use corporate e-mails and avoid using personal e-mails.

The summary could contain a few lines on the results of the sensory analysis.

In the introduction, the difference between hot air drying and microwave drying could be explained in more detail, as the principles of operation are different and are associated with convective and radiative heat transfer mechanisms respectively.

In the freeze-drying experiments, it is necessary to add information on the working pressure used.

In the colour measurements it is necessary to add information on the illiminant and the observer used for the measurements.

It is necessary to add information about the meaning of the variables L*,a* and b*.

In some analyses the equipment used is not mentioned, the format of the equipment used needs to be revised and standardised.

Preferably use dimensionless moisture expressed as moisture rate instead of moisture content in grams of water per gram of sample.

The analysis in figure 1 can be improved by a more in-depth analysis.

In general the discussion should be improved, as it is sometimes too simple and does not contribute to the generation of knowledge in the context of the study.

Author Response

Response to Reviewer 3 Comments

Point 1: Revise the title, perhaps it would be better to talk about microwave pre-treatment or assisted drying.

Response 1: Thanks for the referee's kind advice. We have revised the title, and the detailed revision could be found in the revised version.

Point 2: If possible use corporate e-mails and avoid using personal e-mails.

Response 2: Thanks for the referee’s suggestion. Corresponding author has been changed to corporate e-mails in the revised version.

Point 3: The summary could contain a few lines on the results of the sensory analysis.

Response 3: Thank you very much for your circumspection. We have added some sensory analysis in summary in the revised version. (Please see Line 20-23)

Point 4: In the introduction, the difference between hot air drying and microwave drying could be explained in more detail, as the principles of operation are different and are associated with convective and radiative heat transfer mechanisms respectively.

Response 4: Thank you very much for your circumspection. We have added some describe about microwave irradiation in the Introduction, and the detailed revision could be found in the revised version. (Please see Line 48-51)

Point 5: In the freeze-drying experiments, it is necessary to add information on the working pressure used.

Response 5: Thanks for the referee's kind advice. The information of the freeze-drying experiments have added on the working pressure used in the revised version. (Please see Line 89)

Point 6: In the colour measurements it is necessary to add information on the illiminant and the observer used for the measurements.

It is necessary to add information about the meaning of the variables L*, a* and b*.

Response 6: Thank you very much for your circumspection. The method of analysis of colours has been revised in a revised version, and the detailed revision could be found in line 97-101.

Point 7: In some analyses the equipment used is not mentioned, the format of the equipment used needs to be revised and standardised.

Response 7: Thanks for the referee’s suggestion. We have revised and standardised the format of the equipment, and the detailed revision could be found in the revised version. (Please see Line 81, 88, 92, 96, 111, 119, 122, 126, 143, 167, 175, and 186, et al. )

Point 9: Preferably use dimensionless moisture expressed as moisture rate instead of moisture content in grams of water per gram of sample.

Response 9: Thanks for the referee's kind advice. Figure 1 has been revised in the revised version. (Please see Line 239)

Point 10: The analysis in figure 1 can be improved by a more in-depth analysis.

Response 10: Thanks for the referee's kind advice. We have added some analysis about drying characteristics in the revised version. (Please see Line 232-238)

Point 11: In general the discussion should be improved, as it is sometimes too simple and does not contribute to the generation of knowledge in the context of the study.

Response 11: Thanks for the referee's kind advice. We have improved in the discussion, and the detailed could be found in the revised version.
